# Relative dispersion ratios following fecal microbiota transplant elucidate principles governing microbial migration dynamics

Yadid M. Algavi [1] & Elhanan Borenstein [1,2,3] ✉

Microorganisms frequently migrate from one ecosystem to another. Yet, despite the potential importance of this process in modulating the environment and the microbial ecosystem, our understanding of the fundamental forces that govern microbial dispersion is still lacking. Moreover, while theoretical models and in-vitro experiments have highlighted the contribution of species interactions to community assembly, identifying such interactions in vivo, specifically in communities as complex as the human gut, remains challenging. To address this gap, here we introduce a robust and rigorous computational framework, termed *Relative Dispersion Ratio* (*RDR*) analysis, and leverage data from well-characterized fecal microbiota transplant trials, to rigorously pinpoint dependencies between taxa during the colonization of human gastrointestinal tract. Our analysis identifies numerous pairwise dependencies between co-colonizing microbes during migration between gastrointestinal environments. We further demonstrate that identified dependencies agree with previously reported findings from in-vitro experiments and population-wide distribution patterns. Finally, we explore metabolic dependencies between these taxa and characterize the functional properties that facilitate effective dispersion. Collectively, our findings provide insights into the principles and determinants of community dynamics following ecological translocation, informing potential opportunities for precise community design.

Dispersion, the distribution of taxa from one habitat to another, is a crucial process that allows microbes to colonize new ecosystems, interact with proximate co-existing communities, and transfer genetic material and metabolic byproducts[1]. This fundamental community assembly process shapes the natural environment, from aquatic and terrestrial ecosystems[2–4] to host-associated communities[5]. Within the human microbiome, dispersion is critical for maintaining biodiversity on both spatial (e.g., migration from proximate biogeographic niches through the digestive tract[6]) and temporal (e.g., recurrent waves of invading microbes in the infant gut[7,8]) scales. There is even some evidence that non-communicable diseases, such as cancer and cardiovascular disorders, may be transmitted by microbial components[9], highlighting the importance of understanding the role of migration in community assembly.

From the perspective of the dispersing taxa, to establish a viable population, it has to adapt to the unique conditions in the new habitat, including nutrient availability in this new environment, co-existing competing taxa, and, in host-associated environments, elements of the host immune system[10,11]. Moreover, as bacteria rarely exist in isolation, migration could be portrayed as the mixing of two communities[12]. Indeed, the significance of community-level interactions for the assembly of mixed communities has been

---

[1]Faculty of Medical & Health Sciences, Tel Aviv University, Tel Aviv, Israel. [2]The Blavatnik School of Computer Science, Tel Aviv University, Tel Aviv, Israel. [3]Santa Fe Institute, Santa Fe, NM, USA. ✉e-mail: elbo@tauex.tau.ac.il

highlighted by both mathematical models[13] and experimental studies[14–16]. In the context of the human microbiome, microbial dependencies during dispersion can be studied via naturally occurring host-host interactions. For example, some work has studied the migration of bacteria across human hosts in a shared household[17], during contact sport[18], or via intimate kissing[19]. Notably, however, the scarcity of such data, and the stochastic nature of such interactions, pose various experimental challenges and limit our ability to rigorously study microbial dependencies during translocation. More recently, a new opportunity for examining the dispersion of whole gut microbial populations between hosts has emerged, owing to the increased prevalence of fecal microbiome transplants (FMTs)[20–22]. Indeed, FMT-based data have been utilized to characterize taxa replacement and engraftment efficiency and to explore the medical consequences of such treatments. Yet, while FMT clinical trials facilitate a more controlled and well-defined study setup, the compositional nature of metagenomic data and the statistical challenges it induces, generally limit the ability to infer microbial interactions beyond presence-absence patterns[23]. As a result, many fundamental ecological aspects concerning dispersion in the human gut remained poorly-characterized, such as which taxa are more easily dispersed, what is the role of the potential dependencies between co-migrated taxa, and what causes certain taxa to become dominant in the new host.

In this study, we wish to address these questions by developing a statistically robust measure that we term *Relative Dispersion Ratio (RDR)*, which aims to reveal pairwise interactions between co-colonizing taxa. Using this measure, we analyzed data from a meta-cohort of 8 FMT studies, identifying, cataloging, and characterizing co-colonizing taxa dependencies. We further compared identified interactions with both in-vitro pairwise experiments and population-wide distribution patterns, demonstrating that these interactions play an important role in microbiome assembly. Lastly, we pinpoint the metabolic and functional drivers that may lead to superior dispersion. Overall, our findings provide valuable insights into the mechanisms driving microbial migration and the role of biotic interactions in shaping and transforming the environment.

## Results

### The Relative Dispersion Ratio (RDR)—A robust measure of successful dispersion

In migration scenarios, colonization patterns can provide intriguing insights into microbial assembly dynamics and into the success of colonizing taxa in the new environment and its determinants. To characterize such colonization patterns, we therefore utilized publicly available FMT data. In these data, samples are arranged in triads, with each triad including one donor sample before FMT, and a pair of recipient samples, one before and one after FMT (Fig. 1A). In this work, we refer to each such triad as an *FMT experiment*. In this setting, *colonizers* can be defined as taxa that appeared in both the donor and recipient post-FMT samples but were absent from the recipient pre-FMT sample[24]. Such microbes migrate during FMT from their source environment (the donor's gut), and get established in a new environment (the recipient's gut), in which they were not previously present. In this new host environment, they may face markedly different conditions, community composition, and niche availability, which in turn may determine how abundant they may ultimately become in this host. Our analysis focuses on these taxa, as a vehicle to reveal how microbes adapt and disperse when migrating to a new ecological niche.

Notably, however, due to the compositional nature of metagenomics data, which provide information about the relative abundance of each taxon in the community rather than its absolute abundance, the fold change between a taxon's abundance in the source environment and its abundance in the new environment cannot rigorously attest to its success in this new environment[23]. We therefore introduce below a more robust metric, termed *Relative Dispersion Ratio (RDR)*, which is invariant to compositionality, thus allowing us to uncover complex interactions between colonizing taxa and their ecology more confidently. Specifically, the RDR metric aims to quantify the difference in dispersion between two microbial taxa, and is defined as the ratio of abundances of each of two co-colonizing taxa between the two environments (Fig. 1B). Formally, assuming $a$ and $b$ represent two co-colonizing taxa in some FMT experiment, with $D_a$ and $D_b$ denoting their relative abundances, respectively, in the donor, and $R_a$ and $R_b$ denoting their relative abundances in the post-FMT recipient, then we define *RDR* as $log_{10}[(R_a/R_b)/(D_a/D_b)]$. Focusing on the RDR of pairs of co-

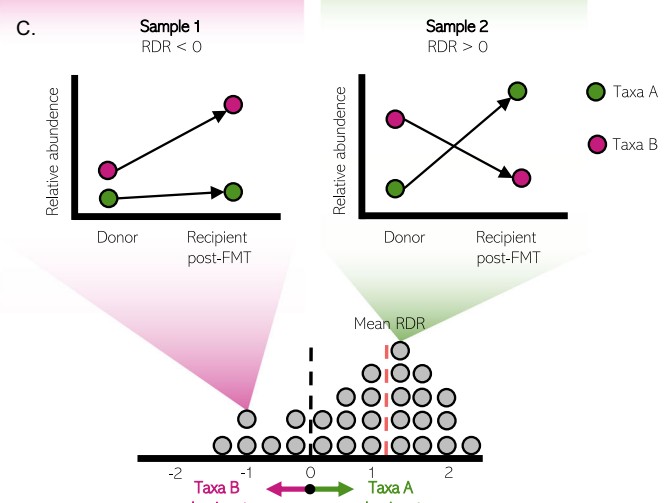

**Fig. 1 | Community structure and dynamics following fecal microbiome transplant treatment. A** Data structure and scale of our FMT dataset collection. Each FMT procedure can be viewed as a mixing of two microbial populations, the donor and recipient pre-FMT, which eventually results in a single recipient post-FMT community. **B** RDR is defined as the fold change of the ratio between the relative abundances of two co-colonizing taxa in the new host (recipient post-FMT) and the former host (the donor). **C** Illustration of possible FMT dynamics as captured by the RDR framework. RDR > 0 indicates relatively better dispersion of taxa A compared to taxa B during migration from the donor to the recipient environments (right top panel), and vice versa for RDR < 0 (left top panel). For each pair of taxa, we can calculate RDR across all FMT experiments in which they appear as co-colonizers, resulting in a distribution of RDR scores (bottom panel). This distribution can be used to calculate the mean RDR (red dashed line), as well as various statistics.

colonizing taxa thus allows us to quantify which bacteria are more dispersive *compared to the other* in the new environment of the recipient host. Put differently, RDR > 0 can be interpreted as indicating a more successful dispersion of the first taxa in compression to the second, while RDR < 0 indicates the opposite (i.e. the second taxa dispersed better than the first; Fig. 1C, top panels). Throughout the text, we will name the better-dispersed taxon in a pair of co-colonizing taxa the *dominant partner*, and the worse-dispersed taxa the *minor partner*.

Note that when RDR > 0, for example, it is still possible that both taxa increased in absolute abundance, that both decreased in absolute abundance, or even that both decreased in relative abundance post-FMT, and that indeed, RDR can only attest to the *relative* success of the two taxa. Importantly, however, since this metric is based on ratios between relative abundances, RDR is not affected by the compositionality of microbiome data and does not suffer from spurious statistical properties as standard analyses of relative abundance data. Moreover, as RDR relies on ratios and accordingly pivots from assessing changes in abundances to examining shifts in the ratios of community members, this approach avoids inaccuracies in determining absolute population shifts and directly analyzes shifts in species dominance through their relative proportions[25,26]. Notably, however, our formulation of RDR is only applicable to considering non-zero abundances, which, as per our definitions above, holds true for any co-colonization event. Finally, it should also be noted that when a pair of taxa appear as co-colonizers in multiple FMT experiments, their RDR can be calculated in each experiment independently, resulting in a distribution of observed RDR scores for these two taxa, as well as their *mean RDR* (Fig. 1C, bottom panel). This distribution can be analyzed, as described below, to determine cases in which observed RDRs are significantly different than zero, suggesting consistent dominance of one of the taxa.

## RDR identifies dispersion differences in FMT settings

We next sought to apply this metric to a large set of FMT samples, aiming to characterize dispersion patterns. To this end, we first constructed a dataset collection obtained from 8 clinical FMT studies. This meta-cohort covers multiple underlying diseases, including three studies of patients with *Clostridium difficile* infections[27–29], three with inflammatory gut disorders[30–32], one with autism spectrum disorder patients[33], and one with melanoma patients[34]. Combined, these datasets thus represent the diversity of the patient populations commonly studied in FMT research, as well as exiting variation in the pertaining clinical procedures (differences in dose, route of administration, and antibiotic preparation). In total, these datasets include 624 16 s rRNA metagenomic samples from 124 patients and 100 healthy donors (Supplementary Table 1). We treated each triad of related samples (donor pre-FMT, recipient pre-FMT, and recipient post-FMT) as an FMT experiment, resulting in a total of 356 such experiments. Although all studies sequenced the V4 or V3-V4 regions, we reanalyzed the raw sequences using QIIME2[35], to reduce variability in downstream analysis. We further profiled each metagenomic sample at the genus level using the GTDB hierarchy[36] and identified 201 unique genus-level taxa (see Methods).

As noted above, to explore how changing environment impacts taxa prevalence, we primarily focused on colonizing taxa. We found that on average each FMT experiment introduces 31 ± 20 new taxa into the recipient ecosystem (Supplementary Fig. 1A), suggesting that colonizing taxa do not disperse in isolation or independently of other taxa. Furthermore, these colonizing taxa form a substantial portion (32 ± 22%) of the post-FMT overall community abundance (Supplementary Fig. 1B). Interestingly, examining the compositions of donor and recipient samples and cataloging all colonizer taxa and their abundances, we further found that 90% of the colonizing taxa undergo on average at least two-fold variation in their relative abundance

following FMT (either two-fold increase or two-fold decrease; Supplementary Fig. 1C), although, as described above, this does not necessarily mean that they increase in absolute abundance.

We finally used the data above to calculate RDR for 14,162 taxa pairs that appear as co-colonizing taxa in at least one FMT experiment. Importantly, however, many of these pairs appear, as co-colonizing taxa, in multiple FMT studies and in multiple FMT experiments (mean 2.2 studies and 7.2 experiments). To focus on pairs of taxa for which the consistency of co-colonization patterns can be quantified confidently, we discarded from our analysis pairs of taxa that were not highly prevalent, considering only the 3160 taxa pairs that appeared as co-colonizing taxa in at least 2 studies and 10 experiments. We then applied a random effects model (accounting for multiple post-FMT collection timepoints and recipient identity) to identify pairs for which the calculated RDR across the various experiments in which they appear is significantly different from 0 (with FDR-corrected *p*-value < 0.1, see Methods for details), indicating that one member of the pair dispersed significantly better than the other. Our analysis identified 374 such pairs, which we will term differentially dispersing pairs throughout the text. For each such pair, we assess the magnitude of differential dispersion by calculating the mean RDR across all experiments in which the pair co-colonized, and the identity of the dominant partner by the sign of the mean RDR (i.e., RDR > 0 indicates that the first species, *a*, is the dominant taxon, and RDR < 0 indicates that the second species, *b*, is the dominant partner; see Fig. 1B). In total, these pairs represent eight phyla and over 81 genera. Figure 2A–D presents the observed colonization patterns for ten representative differently dispersing pairs, including their RDR score in each experiment and mean RDR (Fig. 2A), statistical significance (Fig. 2B), and the number of FMT experiments and studies in which they were found (Fig. 2C, D).

To confirm and demonstrate the generalizability of our findings within the context of FMT studies, we further used a leave-one-study-out cross-validation approach. Specifically, we removed one study from the meta-cohort each time, and for each of the 374 differentially dispersing pairs described above, compared the mean RDR of the pair in this study alone, to the mean RDR calculated for the seven remaining studies. We found that the sign of the mean RDR (and hence the identity of the dominant partner) is the same in >95% of these leave-one-out calculations, indicating that our RDR analysis is not study-specific.

The complete table of mean RDR scores for all differentially dispersing pairs is available in Fig. 2E. Inspecting the identity of the dominant and the minor partners in these pairs, we found that members of the phyla *Bacteroidota* frequently tend to be the dominant partner, followed by *Firmicutes* and *Proteobacteria* that are more often the minor partner (Supplementary Fig. 2, Wilcoxon test). This is in line with previously reported phylum-dependent colonization rates during FMT[20–22]. Among genera with high dominance rates, are prevalent members of the human gut microbiome such as *Bacteroides* and *Akkermansia*, as well as multiple members of the family *Lachnospiraceae*, including *Dorea* and *Blautia*. Indeed, these taxa have been well-documented for their ubiquity in various environments[37–40].

## Comparing RDR scores to results from in-vitro experiments and population-wide distribution patterns

To confirm the dispersion differences inferred by the RDR methodology above, we compared them to findings from in-vitro experiments in which distribution differences between human microbiota members could be measured explicitly. To this end, we obtained data from a study of 36 pairwise in-vitro competition experiments, in which two gut-dwelling strains were incubated in a co-culture for 72 h while their abundances were monitored[41]. While these experimental settings are clearly markedly simpler in comparison to FMT (in-vitro vs in-vivo, co-culture vs community, etc.), we still expect dominant partners to generally gain higher abundance in the co-culture at the end of the

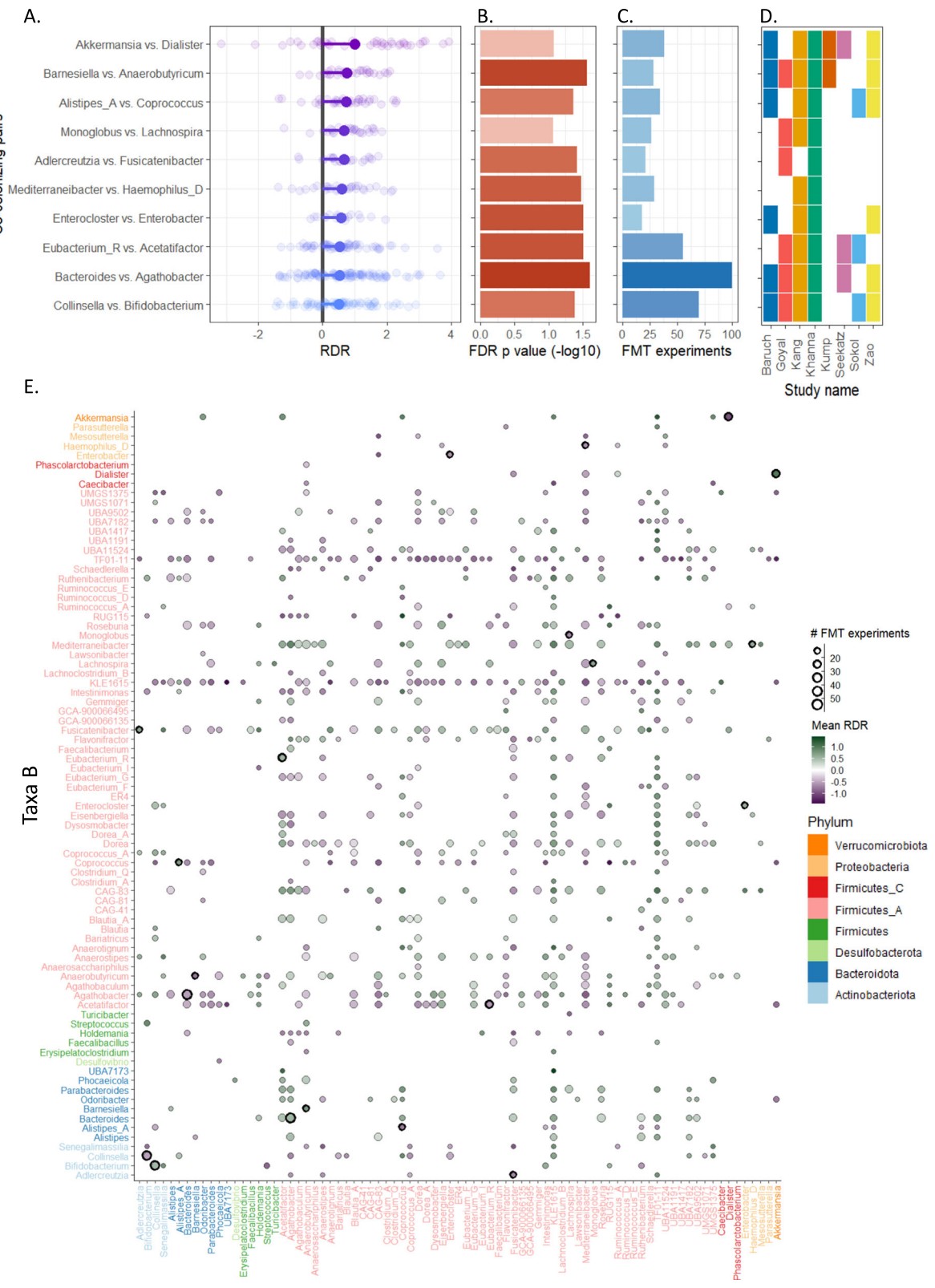

**Fig. 2 | Relative Dispersion Ratio (RDR) scores across differentially dispersing pairs. A** Each dot represents the calculated RDR from a single FMT experiment, with the large dot representing the mean RDR. For convenience, we ordered each pair such that the dominant partner is the first taxa and the minor partner is the second species (and hence, all mean RDR values are positive). **B** FDR-corrected *p*-values obtained by a random effects model. **C** Number of FMT experiments in which the pair co-colonized. **D** Identity and number of studies in which the pair appeared. **E** The complete mean RDR table between all differentially dispersing pairs. The color of each dot indicates the identity of the dominant partner (Green−taxa A on the *X*-axis, Purple−taxa B on the *Y*-axis). Dot size represents the number of FMT experiments in which the pair of taxa co-colonized. The color of the labels on the *X* and *Y*-axis indicates the phylum. Black outer circles denote the taxa pairs included in panel (**A**).

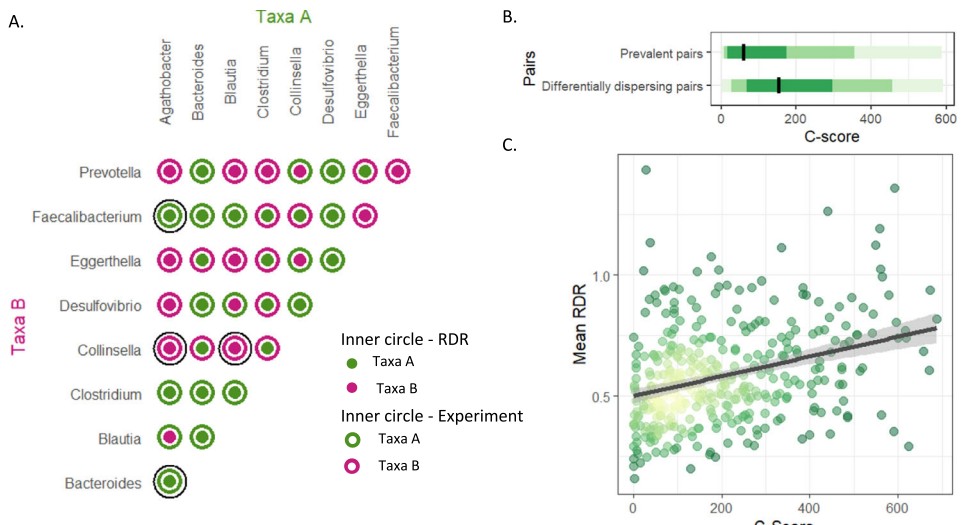

**Fig. 3 | Comparison of RDR to observed patterns in in-vitro experiments and population-wide distributions. A** Comparison between the identity of the dominant taxon as inferred from the RDR framework and from in-vitro competition experiments. Each dot represents a single in-vitro competition experiment between a pair of microbes. The color of the outer and inner circles indicates the identity of the dominant taxa in the in vivo experiment and RDR pairs, respectively (green indicates that the dominant taxon is the one listed on the *x*-axis while purple indicates that the dominant taxon is listed on the *y*-axis). Matching inner and outer colors indicates agreement between the RDR and experiment. The outer-most

black circles highlight the 4 differentially dispersing pairs. **B** The difference in C-score across the American gut project (AGP) cohort between significant RDR pairs and other prevalent pairs (two-sided Wilcoxon signed-rank test, $p = 5.62 * 10^{-30}$). Different shades of green indicate 50%, 80% and 95% confidence intervals. **C** Correlation between C-scores from the AGP cohort and mean RDR scores for all differentially dispersing pairs (Spearman correlation, $\rho = 0.33$, $p = 1.41 * 10^{-10}$, gray line indicates 95% confidence interval, yellow-green point color indicates density).

experiment. Indeed, we found a high degree of agreement between the results of our RDR framework and the in-vitro experiments (Fig. 3A). Specifically, considering the 4 differentially dispersing pairs included in this in-vitro study, we found that all of them agreed on which of the two taxa was dominant. Furthermore, examining all taxa pairs included in this in-vitro assay (i.e., including those for which our analysis did not reach significance and hence were not identified as differentially dispersing pairs), we found that in 24 out of 36 comparisons, the mean RDR and in-vitro experiments again agreed on the dominant taxa (67% of interactions, Cohen's kappa, $\kappa = 0.33$, $p = 0.04$).

To further assess the relevance of our RDR framework for characterizing community assembly, we next examined taxa distribution patterns across human populations. While this setting likely captures additional dimensions of environmental selection beyond differences in dispersion capabilities (such as interactions with other gut-dwelling microbes, host nutrition, and health conditions), it could serve as a proxy for the long-term outcome of such differential dispersion dynamics. To this end, we turned to the American Gut Project (AGP), a cohort containing metagenomic samples from nearly 10,000 participants[42]. To minimize bias due to differences in metagenomic processing pipelines, we re-analyzed the raw sequencing data and annotated it according to the GTDB taxonomy (Methods). We then calculated the checkerboard score (C-score), which is a measure of pairwise taxa distribution across different habitats, characterizing competition among taxa and resulting taxa segregation[43]. We found that differentially dispersing pairs exhibited significantly higher C-scores in comparison to the C-scores of all other prevalent pairs, potentially suggesting the existence of exclusion dynamics between these taxa during community assembly[44,45], which is in line with their differential dispersion success identified via our RDR metric (Fig. 3B, Wilcoxon test $p = 5.62 * 10^{-30}$). Interestingly, we further found that the C-score of these pairs is significantly correlated with their mean RDR ($\rho = 0.33$, $p = 1.41 * 10^{-10}$; spearman correlation test), suggesting that differences in dispersion capabilities are associated with population-wide taxa segregations patterns (Fig. 3C). Taken together, these analyses suggest that the relative dispersion differences inferred via the

RDR framework are well mirrored in both in-vitro experiments and population-wide distribution patterns.

## Establishing RDR dispersion patterns based on mechanistic metabolic model-based methods

We next set out to identify plausible mechanistic determinants of the observed dispersion differences between the two members of each differentially dispersing pair. We first resorted to a metabolic network-based analysis, using specifically a previously introduced framework, termed reverse-ecology[46], which was shown to successfully predict competitive and cooperative interactions between microbial species, as well as ecological design strategies[47,48].

Briefly, for this analysis, we first constructed genus-level metabolic networks, using genomic representation from PICRUST2[49] (Methods). We then applied the seed-set detection algorithm[47,50] to infer the organism's nutritional profile and used the fraction of seed metabolites from the total number of metabolites in the network as a proxy for the nutritional flexibility of this genus and its ability to flourish across multiple biochemical habitats as previously suggested[51]. Comparing this measure to our calculated mean RDR, we found that dominant partners had on average a higher nutritional flexibility (Fig. 4A, two-tailed Wilcoxon test, $p = 9.96 * 10^{-3}$), which can be beneficial for adapting to different metabolite availability in a new environment. We also used the identified seed sets to compute the metabolic competition index between pairs (following Levy et al.[50]). This index was shown to provide a proxy for niche overlap and estimates the potential level of competition one taxon may experience in the presence of the other. Compellingly, we found that the dominant partner experienced significantly less competition in the presence of the minor partner than the minor partner did in the presence of the dominant partner (Fig. 4B). Together, the increased metabolic flexibility and the lower competition may thus indicate an enhanced metabolic capacity of the dominant partner compared to its minor partner.

Finally, we set out to examine whether differences in dispersion success as measured by the RDR framework are also associated with

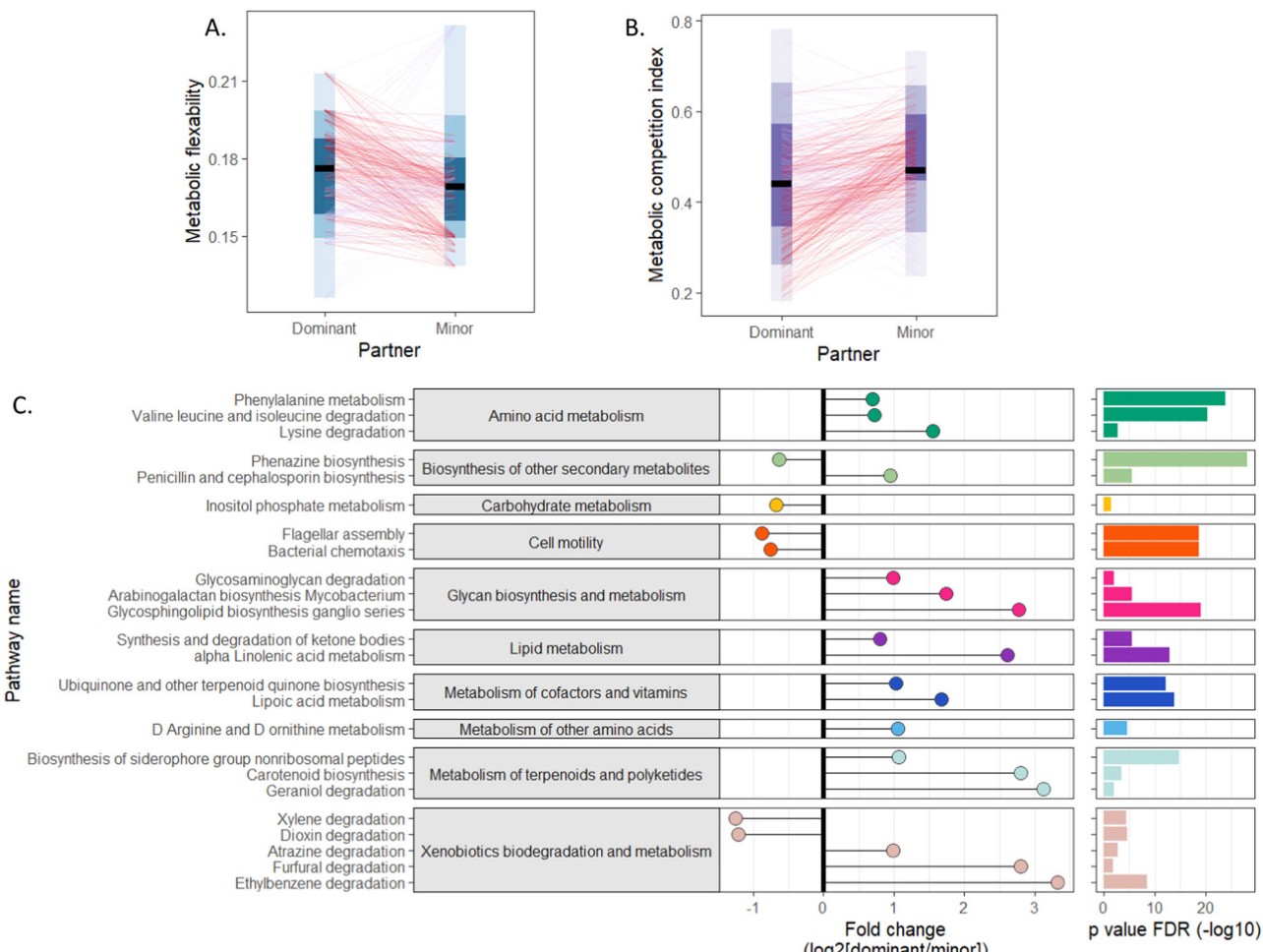

**Fig. 4 | Comparison of the metabolic capacity of the dominant vs. minor partner in differentially dispersing pairs. A** Pairwise comparison of metabolic flexibility (calculated based on the seed-set framework; Methods) between the dominant and minor partner (paired two-sided Wilcoxon signed-rank test, $p = 9.96*10^{-3}$). Different blue shades indicate 50%, 80% and 95% confidence intervals. **B** Pairwise comparison between the metabolic competition index between the dominant and minor partner (paired two-sided Wilcoxon signed-rank test, $p = 1.68*10^{-5}$). Different purple shades indicate 50%, 80% and 95% confidence intervals. **C** Significant differences in bio-synthetic capacities between dominant and minor partners. Pathways are organized according to pathway categories. Positive values indicate enrichment among dominant taxa while negative values indicate depletion (Two-sided Wilcoxon rank-sum test with FDR correction).

differences in biosynthetic and functional capacities between the two members of each differentially dispersing pair. To this end, we compared the number of KEGG KOs in each pathway between the members of each such pair to identify pathways with significantly more or less genes in the dominant vs. the minor partner (Methods). Our analysis identified 24 such KEGG metabolic pathways, spanning multiple functional categories (Fig. 4C). Specifically, this analysis suggested that highly dispersive taxa tend to encode fewer cell motility genes. Indeed, cell motility associated genes are known to have high energetic costs and cause reduced growth rates[52]. In addition, these motility genes have also been associated with lower engraftment rates during FMT[20]. In addition, we found that dominant taxa tend to have a variety of enriched biosynthetic capabilities, including the metabolism of essential amino acids such as valine, lysine, and isoleucine, which have been identified as important factors in the assembly of microbial communities[53]. Dominant taxa were further associated with enrichment of pathways for glycans breakdown, which serve as energy sources for gut microbes. Indeed, the ability to utilize a variety of glycans and dietary fibers has been shown to expand the microbe's ecological niche in a new habitat[54]. Moreover, we observed enrichment in pathways for synthesizing cofactors involved in energy production, such as lipoic acid, used in the TCA cycle, and ubiquinones, redox-active compounds during ATP production[55]. Similarly, bile acid transformation pathways, including those for primary bile acid

and alpha-linolenic acid, were also enriched among dominant taxa, in agreement with the known impact of bile acid repertoire modulation on gut microbiome and host health[56]. Finally, there was an increase in the metabolism of non-ribosomal peptides, which have crucial roles as siderophores and virulence factors[57]. Overall, this analysis has identified several potential functional factors that may contribute to successful dispersion.

**Partners in differentially dispersing pairs are highly informative for predicting post-FMT abundances**

Given the observed consistency of relative dispersion rates in differentially dispersing pairs, and more importantly, the potential metabolic determinants underlying the observed dependencies between co-colonizing taxa, we next examined whether such pairs can also be linked to our ability to predict community structure after FMT. To this end, we used random forest regressor models to predict the relative abundance of every taxon post-FMT based on pre-FMT community compositions in the donor and recipient (Methods). We evaluated the performances of these models using Spearman correlation between the predicted and observed abundance, and defined taxa that exhibit FDR corrected $p$-value $< 0.05$ and Spearman $\rho > 0.3$ as *well-predicted*. Notably, out of the 100 taxa analyzed, 52 were well-predicted with a mean $R^2$ of 0.21 (Fig. 5A). Focusing on these well-predicted taxa, we

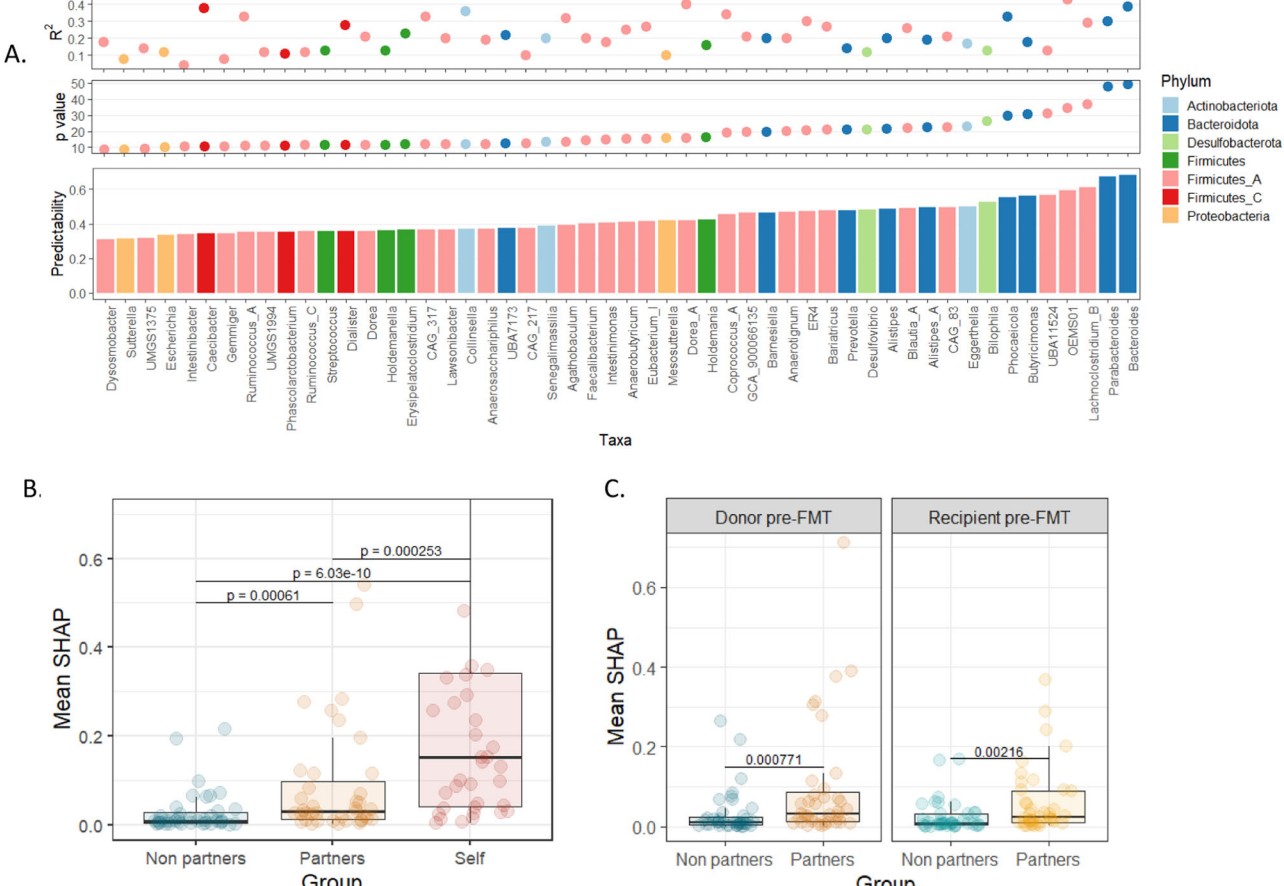

**Fig. 5 | Information gained by co-colonizing taxa for predicting post-FMT community composition. A** Machine learning models predict post-FMT abundance. Bottom panel—predictability (measured by Spearman correlation between observed and predicted abundances); Middle panel—FDR corrected *p*-values (in -log10 scale); Top panel—the model coefficient of determination ($R^2$). As per the GTDB taxonomy, the phylum Firmicutes has now been divided into several distinct phyla. **B** Comparison between SHAP values of 3 different taxa categories: Self (the pre-FMT abundance of the predicted taxa), Partners (partners of the predicted taxa in differentially dispersing pairs), and Non-partners. **C** Comparison between SHAP values (as in Panel B), with separation between donor and recipient profiles. In both panels B and C, significance was calculated using the two-sided Wilcoxon rank-sum test with FDR correction, with $n = 39$ models in each taxa category. The line across each box indicates the median. The whiskers are lines extending from Q1 and Q3 to endpoints that are defined as the most extreme data points within $Q1 - 1.5 \times IQR$ (interquartile range) and $Q3 + 1.5 \times IQR$, respectively.

quantifies the contribution of each pre-FMT taxa in the donor and in the recipient to the model using Shapley additive explanations (SHAP) values[58]. For subsequent analysis, in evaluating each predictive model, we partitioned the pre-FMT taxa into 3 categories: "Self"—the taxa whose abundance post-FMT the model aims to predict, "Partners"—partners of the predicted taxa in differentially dispersing pairs, and "Non-partners" —taxa that are not partner of the predicted taxa in a differentially dispersing pair. Comparing the SHAP contribution of these 3 taxa categories across 39 models that have "partners", we first found, not surprisingly, that the abundance of the predicted taxa ("self") in the pre-FMT samples has the highest contribution to predicting the taxon's post-FMT abundance (Wilcoxon test, Fig. 5B). However, we further found that partners have significantly higher contribution to predicting post-FMT abundance compared to non-partners (Fig. 5B, $p = 6.1*10^{-4}$, FDR corrected). Importantly, this trend holds when considering separately pre-FMT abundances in the donor and in the recipient (Wilcoxon test, Fig. 5C).

## Discussion

Understanding the mechanisms underlying the dispersion of microbial communities is an essential, yet poorly understood ecological domain. Previous studies have stated that inter-microbial interactions between invading taxa play a crucial role in modulating the community in

various habitats, such as soil, marine environments, and plant roots[2–4]. However, studies aiming to characterize such interactions in human-associated communities are still scarce, owing to various statistical and experimental challenges. These difficulties, for example, have promoted multiple previous studies to analyze dispersion across human gut environments based on presence/absence data only[20–22] (*i.e.*, whether the microbe has been detected following FMT or not), ignoring the wealth of information captured in microbiome compositional profiles.

Here, we address some of these challenges and further quantify the outcomes and determinants of microbial dispersion across human gut microbiomes by introducing a new computational framework for analyzing metagenomic data from multiple clinical FMT studies. Our framework aims to uncover consistent differences in dispersion rates, ultimately presenting a web of potential dependencies between migrating gut-dwelling microbes. Importantly, we further provided evidence that identified dispersion differences mirror co-culture experiments and population-wide co-occurrence patterns, and used metabolic modeling and functional enrichment analysis to pinpoint functional and metabolic factors associated with superior dispersion (that were in agreement with reported findings from germ-free mice experiments[59]).

The set of identified co-colonizing taxa pairs that exhibit statistically significant difference in dispersion can be used to better

understand potential microbial dynamics during migration, including, for example, underlying competitive or metabolic interactions. However, while our study represents a step forward in understanding the role of biotic and metabolic interactions in dispersion dynamics, it is important to note that these processes are extremely complex and that multiple other environmental factors, such as nutrient avilability[54,60], host genetic background[61], the activity of the immune system[62], and biogeographic patterns[63], are all likely playing a role in shaping the composition of the community after fecal transplant. Beyond physiological and ecological factors, species dynamics could also be influenced by the clinical protocol. Notably, the lack of standardized FMT protocols may introduce marked variability across trials, each applying a distinct combinations of factors such as the route of administration, antibiotic preparation, and dosage[64]. Hopefully, the increase in the prevalence of multi-omics studies of the gut microbiome and of FMT-based data will allow us to further map these complex processes and the contribution of each of these factors to dispersion dynamics.

Another key limitation of using sequencing-based data for exploring migration dynamics arise from the prevalence of zeros in the metagenomics count data[65,66]. While some of these zeros likely represent real absence of given taxa from a given sample (i.e., biological zeros), many may represent artifacts caused by the stochastic nature of the sequencing procedure or by technical bias (termed sampling zeros and technical zeros, respectively)[67]. Importantly, since FMT procedures specifically aim to eliminate pathogenic taxa and re-introduce missing beneficial taxa, accurately discerning true absence and presence patterns in such data is an essential technical challenge. Unfortunately, however, while multiple computational and statistical methods for addressing this challenge have been proposed in the microbiome literature[68] (as well as in other domains[69]), this is still an open field of inquiry, with no clearly-established or well-accepted best-practice guidelines. Thus, in our work, we have opted for a simplified and more straightforward approach, treating all zeros as true absences. This choice is prevalent among FMT and other microbiome studies, yet, it may clearly introduce some noise into our analyses. Specifically, technical zeros may result in mistakenly labeling true colonizers as environmentally acquired or rejected strains or to label a persistent strain as a colonizers[21]. Such mislabeling may mask or distort observed dispersing patterns, calling for future computational development for refining our results.

Notably, from a border perspective, our research aims to tackle a challenging task, inferring how the functional capabilities of individual microbes collectively impact a given community assembly[70,71]. Previous studies of this topic have proposed bottom-up assembly rules, demonstrating that the interaction between individual taxa can explain, at least partly, the observed patterns in community structure. For example, metabolic network analysis[47] and consumer resource models[47,72] have proposed that competitive forces are the main drivers of the resulting taxonomic profile. Other works[73,74], in contrast, have claimed that cooperative cross-feeding interactions are frequent among microbiome members, especially in resource-limiting niches. In the context of microbial migration, specifically, Diaz-Colunga et al.[75] have described the importance of cohesive inter-microbial interactions between members of soil communities, and highlighted the competitive forces between taxa as driving dispersion differences. Our framework, and other analyses of FMT dynamics are similarly essential for uncovering the forces underlying community assembly in the gut, with important and timely implications for microbiome-based therapy.

More generally, we believe that the framework introduced in this work has the potential to be utilized in multiple other settings, where metagenomic data on the origin and destination of microbial migration are available, and could serve as a rigorous method for comprehensively and systematically analyzing microbial dispersion. Furthermore, while the current formulation of RDR only allows for non-zero values, focusing our analysis and the measured interactions only to co-colonizing taxa, future studies could include pseudo counts, non-parametric approaches, or transformation of the relative abundance values. This, in turn, could expand the framework's capacity to a wider array of interactions, such as the elimination of pathogenic or pro-inflammatory species or cases when donor species fail to colonize. Looking ahead, our findings, in conjunction with those of other studies, offer exciting opportunities for designing tailored microbial communities to meet the challenges in both environmental and clinical contexts.

## Methods

### Data acquisition and preprocessing
We acquired 16 S rRNA amplicon sequencing data from published FMT studies described in Supplementary Table 1. For consistency, we processed and similarly analyzed each dataset. Specifically, we obtained raw fastq files from public repositories (NCBI Sequence Read Archive or European Nucleotide Archive) and processed these data using Qiime2 version 2019-127[35]. We demultiplexed the data using the Qiime2 demux plugin, applied DADA2[76] to denoise the data, and trimmed reads in each dataset to the first position with a median quality score under 30. Next, we applied RDP classifier for taxonomic classification according to the GTDB hierarchy[36]. We further filtered samples with less than 2000 reads or with missing metadata, and similarly removed rare and low abundance taxa, leaving those with abundance >0.05% in at least 5% of the samples. Finally, read counts were normalized to sum to 1 within each sample, resulting in a table of relative abundances.

The obtained FMT data are characterized by triads of samples. Each triad includes one donor sample before FMT, and a pair of recipient samples, one before and one after FMT. Following a previous study that categorized classified taxa into different post-FMT classes[21] when multiple recipient post-FMT samples were available, they were treated as participating in multiple FMT experiments. Based on the observed composition patterns in donor and recipient samples, taxa were defined as "colonizers" if they appeared in both recipient post-FMT and donor samples but were absent from the recipient pre-FMT sample.

### Calculation of relative dispersion ratio
We calculated the Relative Dispersion Ratio (RDR) between every pair of co-colonizing taxa. The RDR is defined according to the following equation:

$$RDR = log_{10}\left(\frac{R_a/R_b}{D_a/D_b}\right) \qquad (1)$$

Where, $R_a$ and $R_b$ are the relative abundances of the co-colonizing taxa $a$ and $b$, respectively, in the post-FMT recipient sample, and $D_a$ and $D_b$ are the abundances of these taxa in the donor pre-FMT sample.

To identify differentially dispersing pairs (i.e. pairs for which the calculate RDR across the various experiments in which they appear is significantly different from 0), we have applied a random effects model. This model considers multiple recipient post-FMT collection timepoints, as well as donor identity (in the few cases where donors are shared among multiple recipients). Formally, the model is represented as:

$$RDR \sim 1 + (1|subject\_id) + (1|donor\_id) \qquad (2)$$

Subsequently, pairs exhibiting a statistically significant intercept were classified as differentially dispersing pairs (FDR $p$-value < 0.1, calculated using the lmerTest package[77]). Additionally, we utilized the intercept as an estimate for the mean RDR (which we term *mean RDR* throughout the text for simplicity).

## Comparison with in-vitro experiments

We collected results from published co-cultured competition experiments[41]. As these experiments were conducted on several strains from the same genus, we used the average completion score across all strains from a given genus. We compared the results of the experiments at the end of the 72 h incubation period to the estimated interactions by RDR. The degree of agreement between the RDR framework and the experimental results was estimated using Cohen's kappa.

## Comparison with population dynamics

Data from the American Gut Project (AGP) was obtained from the original AGP publication's accompanying figshare[42]. To allow integration with the FMT studies, we mapped ASV to GTDB genus-level taxonomy[36]. We then removed low-quality samples and filtered rare and low-abundance taxa, following the same criteria as above. We then calculated C-score between every pair of taxa according to the modified formulation of the original work by Stone and Roberts[43,78], in which the score is standardized by the number of occupied habitats. The following score indicates the degree of segregation between pairs of taxa:

$$c_{ij} = \frac{(r_i - S_{ij})(r_j - S_{ij})}{(r_i + r_j - S_{ij})} \tag{3}$$

Where, $c_{ij}$ is the C-score between taxa $i$ and $j$, $r_i$ and $r_j$ are the number of samples in which the taxa appear and $S_{ij}$ is the number of samples in which the taxa $i$ and $j$ appear together.

## Metabolic network analysis

We used the methods described in Levy et al.[50] to investigate seed-sets and metabolic interactions between pairs of taxa. These techniques have been previously utilized to predict metabolic strategies and relationships between bacteria[47]. Briefly, We employed PICRUST2[49] to estimate genus-level functional profiles. Then, given a list of genes mapped to the Kyoto Encyclopedia of Genes and Genomes (KEGG)[79] orthologous groups (KOs), we inferred the set of metabolic reactions that each genus can perform, using a custom script. This scripts results in a network that consists of nodes representing compounds and edges representing reactions linking substrates to products, and describe the overall metabolic capacities shared between all members of the genus. From this network, we calculated the seed set (using python code from Levy et al.[50]), which reflects the nutritional profile of the taxa, as well as the metabolic competition index, which quantified the similarity between the nutritional profiles of the two taxa.

To compare the functional capacities of taxa, we used KEGG pathway scores based on their genomic content as calculated using the method described previously[80]. These scores represent the number of KEGG orthology groups (KOs) present in each pathway (with KOs associated with multiple pathways being partitioned between these pathways). We focused only on pathways with at least a two-fold difference (i.e., above 200% or below 50%) between the dominant and the minor pairs. For each such pathway we tested differences in scores between the dominant and the minor partners using Wilcoxon test (FDR $p < 0.1$).

## Machine learning analysis

To predict the composition of the post-FMT community, we constructed a machine learning model for each taxon. We specifically used a random forest regressor that receives as features the pre-FMT abundances of all taxa in the recipient and in the donor. We evaluated the models using 10-fold cross validation and estimated performance using Spearman correlation between the observed and predicted post-FMT abundances. Post-FMT taxa with models that exhibit spearman correlation > 0.3 and FDR corrected $p$-value < 0.05 were defined as well-predicted. For these well-predicted taxa, we calculated feature contribution using SHAP values. The machine learning framework was developed using the "Tidymodels" package, random forest models are based on the "Ranger" package[81], and SHAP calculations were performed by the "FastSHAP" package[82].

## Statistics and reproducibility

Statistical calculations were conducted in R (version 4.3.3). We have used the "tidyveres" package for general data handling and cleaning (version 2.00), visualized results using "ggplot2" (version 3.5), and conducted statistical calculations using "vegan" (version 2.6-2). Since this work relies on previously published data, no statistical method was used to predetermine sample size and no data were excluded from the analyses. Similarly, the experiments were not randomized and the investigators were not blinded to allocation during experiments and outcome assessment.

## Reporting summary

Further information on research design is available in the Nature Portfolio Reporting Summary linked to this article.

# Data availability

In this work, we relied on previously published and publicly available datasets. Specifically, for the FMT meta-analysis we used raw-sequencing data from references[27–34]. (these studies are described in detail in supplementary table 1). Data from the American Gut Project (AGP) was obtained from the original AGP publication's accompanying figshare (ref. 42). Lastly, we collected results from published co-cultured competition experiments (ref. 41).

# Code availability

Raw metagenomic data was processed by qiime2 (https://qiime2.org/, version 2019-1), and denoised by DADA2. Similarly, the qiime2 fragment insertion algorithm was used to calculate phylogenetic relationships and PICRUST2 was used to predict functional abundance per ASV. Seed set calculations were conducted using Python code from Levy et al. (ref. 50). Data analysis was conducted in R version 4.02 using the packages "tidyveres" (version 2.00, general data handling and cleaning), "tidymodels" (version 0.14, machine learning pipelines), "ranger" (version 0.13.1, RF model), "FastSHAP" (version 0.1.1, SHAP analysis) and ggplot2 (version 3.5, visualization). Custom analysis code used in this study can be found on GitHub at: https://github.com/borenstein-lab/RDR and on Zenodo (https://doi.org/10.5281/zenodo.11115187).

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

## Acknowledgements

We thank all current and past members of the Borenstein lab for their advice and help. This work was supported in part by NIH Grant U19AG057377 and ISF Grant 2435/19. E.B. is a Faculty Fellow of the Edmond J. Safra Center for Bioinformatics at Tel Aviv University.

## Author contributions

Y.M.A. and E.B. conceived and designed the study and wrote the manuscript. Y.M.A. processed and analyzed the data.

## Competing interests

The authors declare no competing interests.
