## [Peer Review File · Nature Communications]

REVIEWER COMMENTS

Reviewer #1. Post-FMT microbial assembly / Community dynamics. (Remarks to the Author):

The authors describe a relative dispersion ratio (RDR) as a log ratio of relative abundances of two taxa in an FMT donor and recipient. The rationale for developing this index as well as its interpretation and limitations with respect to its relative nature are well described. The study leverages 8 FMT studies for different indications in which the microbiota was characterized by 16S rRNA sequencing to rigorously characterize RDR values and then compares these with in vitro co-culture data and functional inference prediction. The RDR was well supported by the in vitro study and the functional inference highlighted greater metabolic plasticity among dominant colonizers. While I feel that some methods and statements need better support (see specific comments), the context and novelty of this new index are a clear and valuable contribution to the literature.

Specific comments:

Major: Due to the diversity of contemporary research applications for FMT, I think the authors should take 2-3 sentences either when described in results or in methods to comment on the details of the FMT -- dose, preparation (frozen/freeze-dried), route of administration (oral/colonoscopic), time of sample collection, etc.

Minor:

line 112: Can the authors provide some reference or elaborate on the explanation that the ratio is not affected by compositionality?

line 140: What was the time between the before-FMT and after-FMT samples? How variable was this?

Figure 5: Why are Firmicutes listed three times?

line 415: Please elaborate on how multiple post-FMT samples were treated.

lines 429-431: Would selecting a different strain changed the interpretation of the data?

line 452: How was the network constructed? What software?

line 458: The citation refers to a package for shotgun data. How was it used here?

Review of Microbial dispersion in the human gut through the lens of fecal transplant

In this paper, the authors introduced a parameter called Relative Dispersion Ratio (RDR) to describe pairwise interactions between co-colonizing taxa in a fecal matter transplant study. The idea to describe the interactions using relative abundances in recipient's sample relative to that of the donor sample for a pair of taxa is very reasonable. The strength of RDR is that it accounts for compositionality. The paper is well written and illustrated. I have the following comments regarding the RDR measure and its implications.

1. **Zeros in the data:** FMT are carried out by transferring fecal material from a healthy subject to a subject with poor health perhaps caused by gut dysbiosis. Thus, by FMT we may expect to reduce or eliminate pathogenic, and/or opportunistic and/or proinflammatory taxa and gain in the abundance of bacteria that improve gut health. Thus, with post-FMT we expect relative abundances of some taxa in a recipient to go to zero and while others to increase. Researchers would be interested in understanding the interactions among such taxa post-FMT. The definition of RDR requires all relative abundances to be positive and hence cannot be computed to study interactions among such taxa post-FMT. The RDR can only be constructed for taxa which are present in donors as well as post-FMT recipients. Replacing zeros by pseudo-counts may not be reasonable because the zeros are potentially real zeros. Additionally, rare taxa often pose challenges when analyzing microbiome data. Some authors suggest taxon specific bias corrections. I wonder if the authors want to consider those corrections before defining RDR.
2. **Dependencies in the data:** Supplementary Table 1 implies that there could be multiple recipients corresponding to the same donor. This induces dependence in the RDR values among of subject receiving FMT from the same done. These dependencies will affect the variance calculations and any downstream statistical analysis. For example, the p-values reported in Figure 2B could be affected by the underlying dependence.
3. **Within and between subject variability in the data:** Another component of variance that needs to be accounted for, is the variance between and within subjects. Furthermore, since a meta-analyses using 8 cohorts are being performed one needs to take into account the variability between studies. Also, there might be cohort specific biases that need to be accounted for.
4. **Methodology:** I think the statistical methodology and analysis accounting for the above issues is not well-developed or not precisely described.
5. **Terminology:** The authors seem to use "16S" and "metagenomics" interchangeably or I am misreading the paper. Please clarify.

Referee #1

“The authors describe a relative dispersion ratio (RDR) as a log ratio of relative abundances of two taxa in an FMT donor and recipient. The rationale for developing this index as well as its interpretation and limitations with respect to its relative nature are well described. The study leverages 8 FMT studies for different indications in which the microbiota was characterized by 16S rRNA sequencing to rigorously characterize RDR values and then compares these with in vitro co-culture data and functional inference prediction. The RDR was well supported by the in vitro study and the functional inference highlighted greater metabolic plasticity among dominant colonizers. While I feel that some methods and statements need better support (see specific comments), the context and novelty of this new index are a clear and valuable contribution to the literature.”

We thank the reviewer for these encouraging words toward our work and its contribution to the field!

Major Comments

1. *“Due to the diversity of contemporary research applications for FMT, I think the authors should take 2-3 sentences either when described in results or in methods to comment on the details of the FMT -- dose, preparation (frozen/freeze-dried), route of administration (oral/colonoscopic), time of sample collection, etc.”*

We thank the reviewer for raising this important issue. Indeed, FMT is a dynamic and active area of clinical research, characterized by significant variability in procedural approaches. Notably, this issue has been recently addressed in a comprehensive review by Porcari *et al.* (<https://doi.org/10.1016/j.chom.2023.03.020>), finding, for example, that pre-treatment with antibiotics and administration of higher doses of FMT are associated with increased engraftment rates. Conversely, with the mode of administration appears to have only a secondary impact.

While these factors undoubtedly influence the clinical outcomes of FMT, our study primarily explores the ecological interactions between co-colonizing microbial species, and consequently, we believe that procedural variables may be less critical in the context of our specific research focus. Nevertheless, we agree with the reviewer that it is essential to acknowledge this considerable variability in FMT methodologies, alongside other significant confounding factors such as diet, age, and geographic location. These variables, combined with the limited scope and scale of existing studies, are a major challenge for robust statistical analyses in this domain.

Following the reviewer’s comment and to explicitly address this issue, we have now added detailed information about the procedures and characteristics of each study, including dosage, antibiotics preparation, and route of administration, to

Supplementary Table 1 (referred to in line 148). We attach this table below for your convenience.

Supplementary Table 1: FMT datasets analyzed in our study

Study name	Reference	# patients	# samples	# donors	16s region	Dose	Antibiotics preparation	Route of administration	Collection timepoints after FMT (days)	Collection timepoints before
autism_kang_2017	33	18	183	5	V4	High initial dose (2×10^{12} cells) Maintenance dose (2×10^9 cells)	yes	Oral capsules or colonoscopy	7,14,28,56,42,84,98,112	-14
c_diff_khanna_2017	27	33	123	33	V4	homogenized and diluted 50g of fresh stool	no	colonoscopy	7,28	0
c_diff_seekatz_2014	28	10	30	10	V4	25-50ml of stool suspension	yes	Nasogastric	30	-14
c_diff_zao_2018	29	11	43	11	V3-V4	50g of stool suspension	yes	Nasoduodenal	14,28,35,42,70,77,119,126 (subset)	-5
cancer_baruch_2020	34	10	50	2	V4	Colonoscopy: ~50g stool suspension, Capsules: material from ~15g stool (concentrated)	yes	Colonoscopy and oral capsules	7,31,65	-14
crohn_sokol_2020	30	8	56	7	V3-V4	50-100g of stool suspension	no	Colonoscopy	14,42,70,98,126,168	-14
IBD_goyal_2018	31	21	99	21	V4	150g of stool suspension	yes	Colonoscopy	7,30,180	-7
UC_kump_2017	32	13	40	12	V4	50g of stool suspension	yes	Colonoscopy	3,14	-10
Total		124	624	100						

We additionally address this variability and its potential impact on research in this field in the Results and Discussion sections (see lines 144-146 and 385-389).

“Combined, these datasets thus represent the diversity of the patient populations commonly studied in FMT research, as well as exiting variation in the pertaining clinical procedures (differences in dose, route of administration, and antibiotic preparation). “

“Beyond physiological and ecological factors, species dynamics could also be influenced by the clinical protocol. Notably, the lack of standardized FMT protocols may introduce marked variability across trials, each applying a distinct combinations of factors such as the route of administration, antibiotic preparation, and dosage⁶⁴. “

Minor comments

1. *“line 112: Can the authors provide some reference or elaborate on the explanation that the ratio is not affected by compositionality?”*

Thank you again for pointing out this point, which indeed has not been clear enough in the previous version of the paper. Indeed, bypassing the challenges involved in analyzing compositional data is an important feature of the RDR metric and we would like it to be as clear as possible when addressing this in the text. Following the reviewer’s comment, we have extended the explanation of the issue in the Results section and added appropriate citations (lines 114-117):

“Moreover, as RDR relies on ratios and accordingly pivots from assessing changes in abundances to examining shifts in the ratios of community members, this approach avoids inaccuracies in determining absolute population shifts and directly analyzes shifts in species dominance through their relative proportions^{25,26}.”

2. *“line 140: What was the time between the before-FMT and after-FMT samples? How variable was this?”*

The time between the before-FMT and after-FMT is indeed variable and study-specific. Across all studies, before-FMT samples were obtained at 9.75 days on average before FMT (minimum 0 maximum 14 days). Likewise, the first after-FMT sample was obtained within 11.25 days on average after FMT (minimum 3 days, maximum 30 days). Overall, the time between before-FMT and after-FMT sampling is 20.87 days on average with maximum time of 44 days and minimum of 7 days. Donor samples were taken fresh on the day of the transplant. Following your comment, we have added this information to supplementary table 1 (see above). Importantly, in the revised version, we now carefully address the issue of multiple post-FMT samples using a random effects model to account for potential dependencies (see our response to comment #4 below).

3. “Figure 5: Why are Firmicutes listed three times?”

Thank you for this comment, as this point might indeed be confusing to other readers as well. In this work, we used the relatively new Genome Taxonomy Database (GTDB) system which splits the Firmicutes phyla into several phyla. This taxonomical system is based on genomic content and has become very popular since its publication in 2018 (see: <https://doi.org/10.1038/nbt.4229>, with more than 1800 citations up to January 2024).

The use of the GTDB system is described in both the Results and Methods sections (lines 152 and 428, respectively). For clarification, however, we now added a more explicit statement to the legend of Figure 5 (lines 351-352):

“As per the GTDB taxonomy, the phylum Firmicutes has been partitioned into several distinct phyla.”

4. “line 415: Please elaborate on how multiple post-FMT samples were treated.”

Thank you for raising this key point. Indeed, in the original version of the paper, we have analyzed multiple post-FMT samples as independent samples (a practice often used in similar studies). However, following reviewers’ comments, in the revised manuscript, we have re-analyzed all data using a *random effects model* (REM) that accounts for multiple post-FMT samples, as well as multiple recipients corresponding to the same donor. While this new analytical method for determining significant differentially dispersing pairs resulted in various changes throughout the text (e.g., in terms of p-values and effect sizes), it did not qualitatively impact the key findings of our study.

Briefly, the main changes in the results section include: (i) A smaller number of significant RDR pairs, with 374 pairs identified using the REM, more rigorous approach, in comparison to 942 pairs identified in our previous, and more naïve approach. The vast majority of the smaller set (96%) were also included in the larger set. (ii) Since the random effects model allows for a better estimate of effect sizes, we have replaced the mean RDR with the model intercept estimate. Notably, however, these two measures for RDR are very highly correlated (Pearson correlation 0.987, p-value = 10^{-293}).

Following this new approach, the Results section now reads (lines 171 -181):

“We then applied a random effects model (accounting for multiple post-FMT collection timepoints and recipient identity) to identify pairs for which the calculated RDR across the various experiments in which they appear is significantly different from 0 (with FDR-corrected p-value<0.1, see Methods for details), indicating that one member of

the pair dispersed significantly better than the other. Our analysis identified 374 such pairs, which we will term *differentially dispersing pairs* throughout the text. For each such pair, we assess the magnitude of differential dispersion by calculating the mean RDR across all experiments in which the pair co-colonized, and the identity of the dominant partner by the sign of the mean RDR (i.e., $RDR > 0$ indicates that the first species, *a*, is the dominant taxon, and $RDR < 0$ indicates that the second species, *b*, is the dominant partner; see Figure 1B). In total, these pairs represent eight phyla and over 81 genera.”

Similarly, Figure 2 was updated to reflect the change in the set of differentially dispersing pairs.

Finally, we now provide additional details on this random effects model in the Methods section (see lines 448 - 457), which now reads:

“To identify differentially dispersing pairs (i.e. pairs for which the calculate RDR across the various experiments in which they appear is significantly different from 0), we have applied a random effects model. This model considers multiple recipient post-FMT collection timepoints, as well as donor identity (in the few cases where donors are shared among multiple recipients). Formally, the model is represented as:

$$RDR \sim 1 + (1 | subject_id) + (1 | donor_id)$$

Subsequently, pairs exhibiting a statistically significant intercept were classified as differentially dispersing pairs (FDR p-value < 0.1 , calculated using the lmerTest package⁷³). Additionally, we utilized the intercept as an estimate for the mean RDR (which we term *mean RDR* throughout the text for simplicity).”

5. “lines 429-431: Would selecting a different strain changed the interpretation of the data?”

Thanks for pointing out this point. To avoid this potential bias, in the revised manuscript, when multiple strains were used in the experiment, we calculated the *mean* interaction result across all strain in a given genus. This change did not any of our results.

We also modified the Methods section to clarify this point (lines 459 - 461):

“As these experiments were conducted on several strains from the same genus, we used the average completion score across all strains from a given genus.”

6. “line 452: How was the network constructed? What software?”

Thank you for this comment as our original text was indeed unclear. We have edited this explanation to give more information on the network construction and software used (Methods, lines 477-488):

“We used the methods described in Levy *et al.*⁵⁰. to investigate seed-sets and metabolic interactions between pairs of taxa. These techniques have been previously utilized to predict metabolic strategies and relationships between bacteria⁴⁷. Briefly, We employed PICRUST2⁴⁹ to estimate genus-level functional profiles. Then, given a list of genes mapped to the Kyoto Encyclopedia of Genes and Genomes (KEGG)⁷⁵ orthologous groups (KOs), we inferred the set of metabolic reactions that each genus can perform, using a custom script. This script results in a network that consists of nodes representing compounds and edges representing reactions linking substrates to products, and describe the overall metabolic capacities shared between all members of the genus. From this network, we calculated the seed set (using python code from Levy *et al.*⁵⁰), which reflects the nutritional profile of the taxa, as well as the metabolic competition index, which quantified the similarity between the nutritional profiles of the two taxa.”

7. line 458: The citation refers to a package for shotgun data. How was it used here?

Thanks for spotting the error in this citation. We have further verified that indeed all other citations in the manuscript are correct. The corrected citation now reads:

“Manor, O. & Borenstein, E. Revised computational metagenomic processing uncovers hidden and biologically meaningful functional variation in the human microbiome. *Microbiome* **5**, 19 (2017).”

Referee #2:

“In this paper, the authors introduced a parameter called Relative Dispersion Ratio (RDR) to describe pairwise interactions between co-colonizing taxa in a fecal matter transplant study. The idea to describe the interactions using relative abundances in recipient’s sample relative to that of the donor sample for a pair of taxa is very reasonable. The strength of RDR is that it accounts for compositionality. The paper is well written and illustrated. I have the following comments regarding the RDR measure and its implications.”

Thanks for your constructive comments and positive attitude towards our manuscript.

1. *“Zeros in the data: FMT are carried out by transferring fecal material from a healthy subject to a subject with poor health perhaps caused by gut dysbiosis. Thus, by FMT we may expect to reduce or eliminate pathogenic, and/or opportunistic and/or proinflammatory taxa and gain in the abundance of bacteria that improve gut health. Thus, with post-FMT we expect relative abundances of some taxa in a recipient to go to zero and while others to increase. Researchers would be interested in understanding the interactions among such taxa post-FMT. The definition of RDR requires all relative abundances to be positive and hence cannot be computed to study interactions among such taxa post-FMT. The RDR can only be constructed for taxa which are present in donors as well as post-FMT recipients. Replacing zeros by pseudo-counts may not be reasonable because the zeros are potentially real zeros. Additionally, rare taxa often pose challenges when analyzing microbiome data. Some authors suggest taxon specific bias corrections. I wonder if the authors want to consider those corrections before defining RDR.”*

Thank you for presenting this interesting line of thought. To clarify our initial motivation, we have developed the RDR framework specifically to study microbial dispersion, with particular emphasis on the interactions between **co-colonizing** species. As you have correctly mentioned, this metric is only applicable when the abundances are non-zero, but this is in fact always the case for co-colonizers. While you are of course correct that this definition restricts the range of interactions we can measure, focusing on a subset of species pairs, we believe it remains broadly applicable across numerous biologically-relevant migration contexts, even beyond FMT, including for example, migration in soil and deep-sea communities.

Indeed, adjusting RDR to support zero abundance values using pseudocounts or transformation of the relative abundance is a potential extension, and it will expand our capacity to measure a wider array of interactions (such as the elimination of pathogenic or pro-inflammatory species or cases when some donor species fail to colonize). Another option could be the use of non-parametric approaches (such as the worst rank score in epidemiology,

<https://jamanetwork.com/journals/jama/fullarticle/2776315>). However, as the reviewer also noted, such frameworks for correcting zero values may not always be appropriate and may introduce various inaccuracies and biases, requiring extensive validation, benchmarking, and research. Given that the optimal mathematical adjustment is not trivial, we thus opted to focus on a subset of the possible interaction (namely, between co-colonizing species), which is both highly important and support a more rigorous analysis and more confident results.

Yet, to better highlight our definition of RDR (and its applicability) we have adjusted its presentation in the Results section (lines 117-124):

“Notably, however, our formulation of RDR is only applicable to considering non-zero abundances, which, as per our definitions above, holds true for any co-colonization event. Finally, it should also be noted that when a pair of taxa appear as co-colonizers in multiple FMT experiments, their RDR can be calculated in each experiment independently, resulting in a distribution of observed RDR scores for these two taxa, as well as their *mean RDR* (Figure 1C, bottom panel). This distribution can be analyzed, as described below, to determine cases in which observed RDRs are significantly different than zero, suggesting consistent dominance of one of the taxa.”

We have also added a short paragraph to the Discussion section to highlight possible future extension (lines 410-415):

“Furthermore, while the current formulation of RDR only allows for non-zero values, focusing our analysis and the measured interactions only to co-colonizing taxa, future studies could include pseudo counts, non-parametric approaches, or transformation of the relative abundance values. This, in turn, could expand the framework’s capacity to a wider array of interactions, such as the elimination of pathogenic or pro-inflammatory species or cases when donor species fail to colonize.”

1. *“Dependencies in the data: Supplementary Table 1 implies that there could be multiple recipients corresponding to the same donor. This induces dependence in the RDR values among of subject receiving FMT from the same done. These dependencies will affect the variance calculations and any downstream statistical analysis. For example, the p-values reported in Figure 2B could be affected by the underlying dependence.”*
2. *“Within and between subject variability in the data: Another component of variance that needs to be accounted for, is the variance between and within subjects. Furthermore, since a metaanalyses using 8 cohorts are being performed one needs to take into account the variability between studies. Also, there might be cohort specific biases that need to be accounted for.”*
3. *“Methodology: I think the statistical methodology and analysis accounting for the above issues is not well-developed or not precisely described.”*

Thank you for raising this key point. Indeed, in the original version of the paper, we have analyzed multiple post-FMT samples as independent samples (a practice often used in similar studies). However, following reviewers' comments, in the revised manuscript, we have re-analyzed all data using a *random effects model* (REM) that accounts for multiple post-FMT samples, as well as multiple recipients corresponding to the same donor. While this new analytical method for determining significant differentially dispersing pairs resulted in various changes throughout the text (e.g., in terms of p-values and effect sizes), it did not qualitatively impact the key findings of our study.

Briefly, the main changes in the results section include: (i) A smaller number of significant RDR pairs, with 374 pairs identified using the REM, more rigorous approach, in comparison to 942 pairs identified in our previous, and more naïve approach. The vast majority of the smaller set (96%) were also included in the larger set. (ii) Since the random effects model allows for a better estimate of effect sizes, we have replaced the mean RDR with the model intercept estimate. Notably, however, these two measures for RDR are very highly correlated (Pearson correlation 0.987, p-value = 10^{-293}).

Following this new approach, the Results section now reads (line 171-181):

“We then applied a random effects model (accounting for multiple post-FMT collection timepoints and recipient identity) to identify pairs for which the calculated RDR across the various experiments in which they appear is significantly different from 0 (with FDR-corrected p-value < 0.1, see Methods for details), indicating that one member of the pair dispersed significantly better than the other. Our analysis identified 374 such pairs, which we will term *differentially dispersing pairs* throughout the text. For each such pair, we assess the magnitude of differential dispersion by calculating the mean RDR across all experiments in which the pair co-colonized, and the identity of the dominant partner by the sign of the mean RDR (i.e., RDR > 0 indicates that the first species, *a*, is the dominant taxon, and RDR < 0 indicates that the second species, *b*, is the dominant partner; see Figure 1B). In total, these pairs represent eight phyla and over 81 genera.”

Similarly, Figure 2 was updated to reflect the change in the set of differentially dispersing pairs.

Finally, we now provide additional details on this random effects model in the Methods section (see lines 448 - 457), which now reads:

“To identify differentially dispersing pairs (i.e. pairs for which the calculate RDR across the various experiments in which they appear is significantly different from 0), we have applied a random effects model. This model considers multiple recipient post-FMT collection timepoints, as well as donor identity (in the few cases where donors are shared among multiple recipients). Formally, the model is represented as:

$$RDR \sim 1 + (1 | subject_id) + (1 | donor_id)$$

Subsequently, pairs exhibiting a statistically significant intercept were classified as differentially dispersing pairs (FDR p-value <0.1, calculated using the lmerTest package⁷³). Additionally, we utilized the intercept as an estimate for the mean RDR (which we term *mean RDR* throughout the text for simplicity)."

Finally, while the random effects model takes into account donor and recipient identity, it does not account for study identity, owing to several challenges. For example, most pairs appear only in a subset of the 8 studies (mean 4.85 ± 1.6 studies per pair). This highlights the variability in data availability across studies and suggests that the influence of study identity on our findings may be limited. Moreover, it should be noted that while metagenomic meta-analysis studies often have to correct for batch effects that may skewed relative abundance in one study vs. another, here, since we focus on ratios between pairs in the same sample (and in the same study), the impact of batch effects and between-study heterogeneity may not be as crucial.

4. *Terminology: The authors seem to use "16S" and "metagenomics" interchangeably or I am misreading the paper. Please clarify.*

In this work we have analyzed 16s rRNA sequencing data as it was (and still is) the most prevalent data on FMT studies. In our writing, we use the term metagenomics in its broader sense, which refers to a culture-free study of mixed community samples and could thus refer to both 16s and shotgun sequencing. This usage of the term metagenomic is quite common in the microbiome domain (see, for example in this highly cited review <https://www.nature.com/articles/s41579-018-0029-9>).

REVIEWERS' COMMENTS

Reviewer #1 (Remarks to the Author):

The authors have successfully addressed all of my comments.

Reviewer #2 (Remarks to the Author):

I thank the authors for addressing all my comments as well as other reviewer's comments. While I am generally satisfied with the revision, I am still concerned about my question regarding zeros. In a way, the purpose of FMTs is to fill gaps (i.e., enrich) the microbial ecology of unhealthy gut using specimens from a healthy gut. Thus, in principle we are interested in those structural zero in unhealthy population shifted to non-zero values. Conversely, the microbes introduced from a healthy sample may displace some of the opportunistic (perhaps pathogenic) bacteria in an unhealthy gut which now become zero or rare after FMT. The authors' methodology ignores all such taxa and focuses only on those that co-exist under both conditions. Furthermore, as I noted in my previous report, low abundance rare taxa may result in zero counts not because they are structural zeros, but they are zeros due to sampling depth, which need to be treated differently. Even if the authors limit their measure to "co-colonizing" taxa, the rare taxa will pose a problem because they may be zeros in some samples and not others, due to sampling depth. In such instances, I am guessing the authors propose to only complete data. As demonstrated in the literature, adding pseudo-counts is not a good solution for these problems. Hence, as defined in the paper, the RDR measure ignores some of the important pairwise interactions. I acknowledge that zeros are a major problem with most data analytic pipelines. I therefore would have liked to see a simulation study where the authors include some zeros, structural as well as zeros due to rare taxa and see how their measure performs. It will be particularly interesting to see the performance under different levels of rarity and sample sizes.

*Additional confidential comments from Reviewer #2:

Yes, zeros are not satisfactorily addressed in the literature, and it is not unique to this paper or to these authors. It is a technical challenge.

Having said that, ironically, FMTs are a way to deal with "missing healthy" microbes from the gut of a patient and conversely, reduce or eliminate opportunistic pathogens from the disease subject's gut

because the gut is being seeded by “healthy” taxa from a healthy donor. Thus, in my view, the zero counts are specifically important in the context of FMT unlike in the other contexts. Their methodology ignores all such taxa.

In lieu of an extensive simulation studies that could potentially slow down the publication of this paper, perhaps you could request the authors to have an extensive discussion on this issue in their discussion section. It should not take too much time/effort for the authors to write a paragraph along these lines and it will improve the quality of the paper and may even offer opportunities for future methods papers on this subject.

In the end, I am totally fine with your decision because in principle it is a good paper. I want to be helpful to the authors as well as the review process. Hope you find the above remark useful.

May 6, 2024

The specific comments of Referee 2 are carefully addressed below.

Referee #2

I thank the authors for addressing all my comments as well as other reviewer's comments. While I am generally satisfied with the revision, I am still concerned about my question regarding zeros. In a way, the purpose of FMTs is to fill gaps (i.e., enrich) the microbial ecology of unhealthy gut using specimens from a healthy gut. Thus, in principle we are interested in those structural zero in unhealthy population shifted to non-zero values. Conversely, the microbes introduced from a healthy sample may displace some of the opportunistic (perhaps pathogenic) bacteria in an unhealthy gut which now become zero or rare after FMT. The authors' methodology ignores all such taxa and focuses only on those that co-exist under both conditions. Furthermore, as I noted in my previous report, low abundance rare taxa may result in zero counts not because they are structural zeros, but they are zeros due to sampling depth, which need to be treated differently. Even if the authors limit their measure to "co-colonizing" taxa, the rare taxa will pose a problem because they may be zeros in some samples and not others, due to sampling depth. In such instances, I am guessing the authors propose to only complete data. As demonstrated in the literature, adding pseudo-counts is not a good solution for these problems. Hence, as defined in the paper, the RDR measure ignores some of the important pairwise interactions. I acknowledge that zeros are a major problem with most data analytic pipelines. I therefore would have liked to see a simulation study where the authors include some zeros, structural as well as zeros due to rare taxa and see how their measure performs. It will be particularly interesting to see the performance under different levels of rarity and sample sizes.

We thank the reviewer for this insight. Following this comment and the guidelines we received from the editor, we have added a comprehensive discussion of this limitation in the Discussion section (see lines 339-355).

Elhanan Borenstein, Ph.D.

Professor, Blavatnik School of Computer Science, Tel Aviv University
Professor, Faculty of Medical & Health Sciences, Tel Aviv University
Edmond J. Safra Center for Bioinformatics, Tel Aviv University
Affiliate Professor, Dept' of Genome Sciences, University of Washington
External Professor, Santa Fe Institute